# DEEP ACTIVE LEARNING WITH NOISE STABILITY

## ABSTRACT

Uncertainty estimation for unlabeled data is crucial to active learning. With a deep neural network employed as the backbone model, the data selection process is highly challenged due to the potential over-confidence of the model inference. Existing methods usually resort to multi-pass model training or adversarial training to handle this challenge. This tends to result in complex and inefficient pipelines, which would render the methods impractical. In this work, we propose a novel Single-Training Multi-Inference algorithm that leverages noise stability to estimate data uncertainty. A key idea behind is a magnitude measurement of output derivation from the original observation when the model parameters are randomly perturbed by noise. We provide analysis by using the small Gaussian noise theory and show that our method has a strong connection with the classical theory of variance reduction. That is, by labelling a data sample of higher uncertainty, as indicated by the inverse noise stability, our method contributes more to reducing the variance of existing training samples. Despite its simplicity, our method outperforms the state-of-the-art active learning baselines in image classification and semantic segmentation tasks.

## 1 INTRODUCTION

The success of training a deep neural network highly depends on a huge amount of labeled data. Nevertheless, in the era of big data, much data is unlabeled. This imposes a challenge on exploiting appropriate learning fashions. For instance, given a pool of unlabeled data, one can employ unsupervised manner to learn a model. However, the performance of unsupervised or semi-supervised learning is still bound to that of supervised learning (Yoo & Kweon, 2019). An ideal option could be annotating all the given data, so as to use the supervised way to learn a model. Nevertheless, the cost for data annotation could be extremely high, such as in medical area (Gorriz et al., 2017; Konyushkova et al., 2017). Due to the limited labeling budget, it may be more feasible to annotate a portion of the given data. This is also aligned with the prerequisite of semi-supervised learning. Then one can ask: which data is deserved for being annotated?

Active learning was proposed to solve this question. One of the main ideas, namely *uncertainty-based method*, aims to select a portion of the most uncertain or informative data from an unlabeled pool for annotation. The newly annotated data are then used to train a task model. Note that uncertainty estimation with deep neural networks (DNNs) remains challenging, due to the potential over-confidence of deep models. That is, the primitive softmax output[1] as a score does not necessarily reflect a reliable uncertainty or confidence for the prediction. Several methods have been proposed to address this challenge. For example, MC-Dropout (Gal & Ghahramani, 2016) implements a Single-Training Multi-Inference algorithm by aggregating multiple confidences sampled by Monte Carlo dropout. The algorithm has been shown to be equivalent to the approximate Bayesian inference. Another typical idea is query-by-committee (Freund et al., 1997; Gorriz et al., 2017), or QBC in short, which requires training multiple auxiliary models as a committee. Besides, many other efforts suggest to use specially designed auxiliary modules or training fashions, such as variational auto-encoder (Sinha et al., 2019), adversarial learning (Ducoffe & Precioso, 2018; Mayer & Timofte, 2020) and Graph Convolutional Network (Caramalau et al., 2021).

In this work, we study the problem of deep active learning from a different viewpoint, inspired by a relevant topic of *noise stability*. Noise stability, in terms of the output stability w.r.t. the input

---

[1]Considering classification problems

noise, is well studied in literature. For example, the authors in (Bishop, 1995) propose to train a model with perturbed input samples as a regularizer to improve the model performance. A recent theoretical work (Arora et al., 2018) shows that, the stability of each layer's computation to noise injected at lower layers acts as a good indicator of the generalization bounds for DNNs. Behind these achievements is an intuitive implication that, a model (or more exactly, its parameters) robust to perturbed input examples tends to be easy to recognize unseen examples. For the active learning problem, we give an analogous intuition by exchanging the role of the input and parameter that, an example robust to the perturbed current parameters tends to be easy to be recognized by a future model. In other words, *examples less robust to the parameter perturbation would be regarded as having higher uncertainty.*

Specifically, we introduce a simple algorithm of uncertainty estimation by measuring how far does the output deviate from the original value, when imposing a small noise on the parameters. The induced distance is used as a measurement of uncertainty used to select unlabeled examples in active learning. We provide a theoretical analysis under the condition that the noise magnitude is small. By imposing standard multivariate Gaussian noise to the parameter, we prove that the proposed noise stability is equivalent to the parameter-output Jacobian norm under the first order Taylor assumption. Furthermore, we show that, our algorithm has a close connection with variance reduction (Cohn et al., 1996). That is, selecting unlabeled samples with low noise stability would yield the same effect of reducing the prediction variance of existing training samples.

Our method is easy to implement and free of customized auxiliary models. Therefore, it can be exploited in various tasks, such as image classification and semantic segmentation, leading to its task-agnostic nature. We conduct extensive experiments to evaluate our method on various datasets including Cifar10 (Krizhevsky et al., 2009), Cifar100 (Krizhevsky et al., 2009), SVHN (Netzer et al., 2011), Caltech101 (Fei-Fei et al., 2006), Cityscapes (Cordts et al., 2016), and cryo-ET (Chen et al., 2017). The performance of our method significantly exceeds that of the state-of-the-art.

The contributions of this work are summarized as follows.

1. We propose a novel effective method of noise stability to select unlabeled data for active learning. The proposed method is free of any auxiliary models or special training fashions.

2. We provide a theoretical analysis to show that, selecting unlabeled data with higher noise stability is equivalent to selecting that with higher Jacobian norm w.r.t. the parameter. We also establish a connection between noise stability and variance reduction for the existing training set.

3. We conduct extensive experiments on benchmark datasets of image classification, semantic segmentation, and 3D cryo-ET subtomogram classification. The results demonstrate the effectiveness of the proposed method.

## 2    ACTIVE LEARNING WITH NOISE STABILITY

### 2.1    PROBLEM DEFINITION

Here we formulate the active learning problem. Given a pool of unlabeled data $\{X_U\}$, and a labeled pool $\{X_L, Y_L\}$ which is initially empty. Active learning aims to select a portion of data from $\{X_U\}$ depending on labeling budget (e.g. select 2500 samples out of 50000 at a time). The selected data $X_N$ is then annotated by human oracles (or equivalent), and added to the labeled pool. That is, $\{X_L, Y_L\} \leftarrow \{X_L, Y_L\} + \{X_N, Y_N\}$, where $Y_N$ is the new annotation for $X_N$. The unlabeled pool is then updated by removing the selected data: $\{X_U\} \leftarrow \{X_U\} - \{X_N\}$. $\{X_L, Y_L\}$ is then used to train a task model $f(.; \theta)$ in supervised fashion by minimizing an empirical loss $L$ (e.g. cross-entropy loss for classification), where $f(.; \theta)$ denotes a neural network $f(.)$ parameterized with $\theta$. Note that in this paper,

### 2.2    UNCERTAINTY ESTIMATION WITH NOISE STABILITY

We use the noise stability to estimate uncertainty for unlabeled data. Specifically, for an input sample $x$, we explore to what degree will the model's output deviate from the original observation $f(x; \theta)$, when adding random noise to the model parameter $\theta$. Let $\xi \Delta \theta$ denote the added noise,

---

**Algorithm 1:** Active learning with noise stability as uncertainty estimation.

---

**Input** : $\mathcal{T}$: random initialized task model, $\mathcal{U}$: unlabeled pool of training data, $\mathcal{L}$: labeled pool of training data, $\mathcal{C}$: number of cycles in active learning;

**Output:** $\boldsymbol{\theta}$: Final learned parameter of $\mathcal{T}$;

1 **begin**
2    **for** $i \leftarrow 1$ **to** $\mathcal{C}$ **do**
3       train $\mathcal{T}$ with $\mathcal{L}$, obtaining the current parameter $\boldsymbol{\theta}$;
4       **for** $k \leftarrow 1$ **to** $K$ **do**
5          sample $\Delta\boldsymbol{\theta} \sim \mathcal{N}(\mathbf{0}, \sigma^2\mathbf{I})$;
6          create a perturbed model $\mathcal{T}'$ parameterized with $\boldsymbol{\theta} + \xi\Delta\boldsymbol{\theta}$;
7          **for** *every* $\boldsymbol{x}$ *in* $\mathcal{U}$ **do**
8             calculate $R^{(k)}(\boldsymbol{x}) = \|\mathcal{T}(\boldsymbol{x}) - \mathcal{T}'(\boldsymbol{x})\|^2$;
9       compute every $R(\boldsymbol{x})$ by averaging all $R^{(k)}(\boldsymbol{x})$;
10      select samples in $\mathcal{U}$ with top $N$ largest $R(\boldsymbol{x})$ as $\{X_N\}$, and obtain their labels $\{Y_N\}$;
11      update $\mathcal{L}$ with $\mathcal{L} = \mathcal{L} \bigcup \{X_N, Y_N\}$;
12      update $\mathcal{U}$ with $\mathcal{U} = \mathcal{U} \setminus \{X_N\}$;
13    return $\boldsymbol{\theta}$;

---

where $\xi$ controls the magnitude of the noise. If $\Delta\boldsymbol{\theta}$ conforms to the standard multivariate Gaussian distribution[2] $\mathcal{N}(\mathbf{0}, \sigma^2\mathbf{I})$, the criterion to quantify the degree of noise *instability* can be formulated as

$$R(\boldsymbol{x}) = \mathop{\mathbb{E}}_{\Delta\boldsymbol{\theta}\sim\mathcal{N}(\mathbf{0},\sigma^2\mathbf{I})} \|f(\boldsymbol{x};\boldsymbol{\theta}) - f(\boldsymbol{x};\boldsymbol{\theta} + \xi\Delta\boldsymbol{\theta})\|^2, \tag{1}$$

where $\|.\|^2$ is the $L^2$ norm of a vector. In active learning, we select samples with largest $R(\boldsymbol{x})$ for annotation. To approximate the expectation, we perform a Monto-carlo sampling from the multivariate Guassian distribution to generate multiple $\Delta\boldsymbol{\theta}^{(i)}$. Then we get the approximated noise stability by

$$R(\boldsymbol{x}) = \frac{1}{K} \sum_{i=1}^{K} \|f(\boldsymbol{x};\boldsymbol{\theta}) - f(\boldsymbol{x};\boldsymbol{\theta} + \xi^{(i)}\Delta\boldsymbol{\theta}^{(i)})\|^2. \tag{2}$$

The noise magnitude should be small relative to that of the original parameter (e.g. $\xi^{(i)} = 10^{-3}\frac{\|\boldsymbol{\theta}\|}{\|\Delta\boldsymbol{\theta}^{(i)}\|}$) in order to avoid catastrophic perturbation to the clean model. Note that, we use a single $\xi^{(i)}$ for the entire $\Delta\boldsymbol{\theta}^{(i)}$, i.e. each element in $\Delta\boldsymbol{\theta}^{(i)}$ is equally re-scaled by $\xi^{(i)}$. As for the sampling number, we find the setting $K = 5$ or $10$ works well in practice. A complete procedure of our method is presented in Algorithm 1.

## 3 THEORETICAL UNDERSTANDINGS

In this section, we provide theoretical understandings about our simple method for uncertainty estimation. Our main conclusions are summarized as follows.

- When the noise magnitude $\xi$ is sufficiently small, selecting data according to $R(\boldsymbol{x})$ in Eq (1) is equivalent to selecting data by the Frobenius norm of parameter-output Jacobian w.r.t. $f$.
- Based on the above conclusion, selecting new samples with higher $R(\boldsymbol{x})$ are expected to contribute more in reducing the prediction variance of existing training samples.

### 3.1 NOISE STABILITY AS JACOBIAN NORM

Let $f(\boldsymbol{x};\boldsymbol{\theta})$ be differentiable (w.r.t. $\boldsymbol{\theta}$) at the point $\boldsymbol{\theta}$ given $\boldsymbol{x}$ as the input. When the imposed noise $\xi\Delta\boldsymbol{\theta}$ in Eq (1) has a sufficiently small magnitude, we use the first-order Taylor expansion to estimate $f(\boldsymbol{x};\boldsymbol{\theta} + \xi\Delta\boldsymbol{\theta})$ as

$$f(\boldsymbol{x};\boldsymbol{\theta} + \xi\Delta\boldsymbol{\theta}) \approx f(\boldsymbol{x};\boldsymbol{\theta}) + \mathrm{J}_{\boldsymbol{\theta}}(\boldsymbol{x};\boldsymbol{\theta})\xi\Delta\boldsymbol{\theta}, \tag{3}$$

---

[2]See Appendix A.1 for preliminaries about the standard multivariate Gaussian distribution

where $J_{\boldsymbol{\theta}}(\boldsymbol{x};\boldsymbol{\theta})$ is the Jacobian matrix of $f$ with respect to the parameter $\boldsymbol{\theta}$ as $J_{\boldsymbol{\theta}}(\boldsymbol{x};\boldsymbol{\theta})_{(i,j)} = \partial f(\boldsymbol{x};\boldsymbol{\theta})_{(i)}/\partial \boldsymbol{\theta}_{(j)}$. By substituting the Taylor approximation into Eq (1), the uncertainty estimation can be reformulated as

$$R(\boldsymbol{x}) = \mathop{\mathbb{E}}_{\Delta\boldsymbol{\theta}\sim\mathcal{N}(\mathbf{0},\sigma^2\mathbf{I})} \|J_{\boldsymbol{\theta}}(\boldsymbol{x};\boldsymbol{\theta})\xi\Delta\boldsymbol{\theta}\|^2. \tag{4}$$

Since the elements of $\Delta\boldsymbol{\theta}$ are independent with each other, it's easy to derive that the expected uncertainty in Eq (4) is in proportion with the Frobenius norm of Jacobian as

$$R(\boldsymbol{x}) = \xi^2\sigma^2\|J_{\boldsymbol{\theta}}(\boldsymbol{x};\boldsymbol{\theta})\|_F^2. \tag{5}$$

In fact, Eq (5) stands for any appropriate noise distribution with the above mentioned independence prerequisite and zero mean. *We put a detailed proof of Eq (5) in Appendix A.2.*

## 3.2 CONNECTIONS WITH PREDICTION VARIANCE REDUCTION

We next show an interesting connection between uncertainty estimation with noise stability and existing theoretical literature of variance reduction. For convenience, we assume that $\hat{y} = f(.;\boldsymbol{\theta})$ is a real-valued function parameterized with $\boldsymbol{\theta}$ that results in $\hat{y} : \mathbb{R}^d \rightarrow \mathbb{R}$, and the results are easy to generalize to vector-valued functions. Given a labeled training set $\mathcal{L} = \{\boldsymbol{x}_i, y_i\}_{i=1}^m$, we aim at minimizing the Mean Squared Error as

$$S^2 = \frac{1}{m}\sum_{i=1}^{m}(f(\boldsymbol{x}_i;\boldsymbol{\theta}) - y_i)^2. \tag{6}$$

Then, we define by $\mathbf{J}_{\hat{y}}(\boldsymbol{x})$ the Jacobian w.r.t. the parameter $\boldsymbol{\theta}$ at $\boldsymbol{x}$, i.e. $\partial\hat{y}(\boldsymbol{x})/\partial\boldsymbol{\theta}$. Let $\mathbf{A}_{S^2}$ denote the Fisher Information Matrix that $\mathbf{A}_{S^2} = \frac{1}{S^2}\partial^2 S^2/\partial\boldsymbol{\theta}^2$. Note that we omit the notation of $\boldsymbol{\theta}$ in $\mathbf{J}$ and $\mathbf{A}$ to without ambiguity. According to the well-established theory of variance estimation (Thisted, 1988), the output variance at the input point $\boldsymbol{x}$ can be represented as

$$\sigma_{\hat{y}}^2(\boldsymbol{x}) \approx J_{\boldsymbol{\theta}}(\boldsymbol{x};\boldsymbol{\theta})^T\mathbf{A}_{S^2}^{-1}J_{\boldsymbol{\theta}}(\boldsymbol{x};\boldsymbol{\theta}). \tag{7}$$

With the normality and local linearity assumptions, the authors in (MacKay, 1992; Cohn, 1996) further quantify the change of output variance. Specifically, when a new sample $\tilde{\boldsymbol{x}}$ is labeled and added to the training set, the expected new output variance at $\boldsymbol{x}$ is

$$\tilde{\sigma}_{\hat{y}}^2(\boldsymbol{x}) \approx \sigma_{\hat{y}}^2(\boldsymbol{x}) - \frac{\sigma_{\hat{y}}^2(\boldsymbol{x}, \tilde{\boldsymbol{x}})}{S^2 + \sigma_{\hat{y}}^2(\tilde{\boldsymbol{x}})}, \tag{8}$$

where $\sigma_{\hat{y}}^2(\boldsymbol{x}, \tilde{\boldsymbol{x}})$ is defined as

$$\sigma_{\hat{y}}(\boldsymbol{x}, \tilde{\boldsymbol{x}}) \equiv J_{\boldsymbol{\theta}}(\boldsymbol{x};\boldsymbol{\theta})^T\mathbf{A}_{S^2}^{-1}J_{\boldsymbol{\theta}}(\tilde{\boldsymbol{x}};\boldsymbol{\theta}). \tag{9}$$

Therefore, by adding a new sample $\tilde{\boldsymbol{x}}$, the reduction of output variance $\sigma_{\hat{y}}^2(\boldsymbol{x})$ is:

$$
\begin{aligned}
\mathop{\Delta}_{\tilde{\boldsymbol{x}}}\sigma_{\hat{y}}^2(\boldsymbol{x}) &= \frac{\sigma_{\hat{y}}^2(\boldsymbol{x}, \tilde{\boldsymbol{x}})}{S^2 + \sigma_{\hat{y}}^2(\tilde{\boldsymbol{x}})} \\
&= \frac{\|J_{\boldsymbol{\theta}}(\boldsymbol{x};\boldsymbol{\theta})^T\mathbf{A}_{S^2}^{-1}J_{\boldsymbol{\theta}}(\tilde{\boldsymbol{x}};\boldsymbol{\theta})\|^2}{S^2 + J_{\boldsymbol{\theta}}(\tilde{\boldsymbol{x}};\boldsymbol{\theta})^T\mathbf{A}_{S^2}^{-1}J_{\boldsymbol{\theta}}(\tilde{\boldsymbol{x}};\boldsymbol{\theta})} \\
&= \frac{\|J_{\boldsymbol{\theta}}(\boldsymbol{x};\boldsymbol{\theta})^T\mathbf{A}_{S^2}^{-1}\boldsymbol{u}(\tilde{\boldsymbol{x}})\|^2 \cdot \|J_{\boldsymbol{\theta}}(\tilde{\boldsymbol{x}};\boldsymbol{\theta})\|^2}{S^2 + \|\boldsymbol{u}(\tilde{\boldsymbol{x}})^T\mathbf{A}_{S^2}^{-1}\boldsymbol{u}(\tilde{\boldsymbol{x}})\| \cdot \|J_{\boldsymbol{\theta}}(\tilde{\boldsymbol{x}};\boldsymbol{\theta})\|^2},
\end{aligned}
\tag{10}
$$

where we use $\boldsymbol{u}(\tilde{\boldsymbol{x}})$ with the unit length to denote the direction of $\mathbf{J}_{\hat{y}}(\tilde{\boldsymbol{x}})$ as $\boldsymbol{u}(\tilde{\boldsymbol{x}}) = \frac{J_{\boldsymbol{\theta}}(\tilde{\boldsymbol{x}};\boldsymbol{\theta})}{\|J_{\boldsymbol{\theta}}(\tilde{\boldsymbol{x}};\boldsymbol{\theta})\|^2}$. Eq (10) implies that, for any direction $\boldsymbol{u}(\tilde{\boldsymbol{x}})$ the optimal choice for maximizing $\mathop{\Delta}_{\tilde{\boldsymbol{x}}}\sigma_{\hat{y}}^2(\boldsymbol{x})$ is the ones with the largest $\|J_{\boldsymbol{\theta}}(\tilde{\boldsymbol{x}};\boldsymbol{\theta})\|^2$. Though the objective is also affected by the direction $\boldsymbol{u}(\tilde{\boldsymbol{x}})$, searching for an optimal direction is usually inefficient due to the characteristic of almost orthogonality of high-dimensional independent vectors (Vershynin, 2018).

Active learning with the objective of maximizing variance reduction has been successfully validated for shallow models such as mixture of Gaussians and kernel regression (Cohn et al., 1996). However, computing the inversion of $\mathbf{A}$ for each new sample is too cumbersome for deep neural networks. Our work establishes a strong connection between variance reduction and noise stability, and renders the algorithm implementable easily.

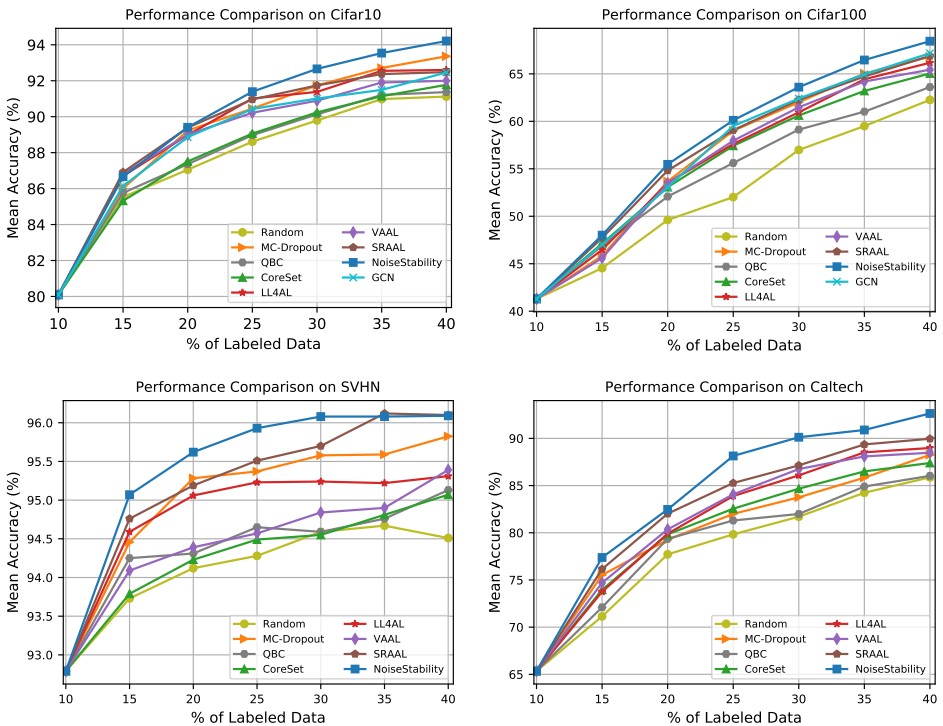

Figure 1: Classification performance due to different active learning methods on Cifar10 (**top left**), Cifar100 (**top right**), SVHN (**bottom left**), and Caltech101 (**bottom right**).

## 4 EXPERIMENTAL RESULTS

We compare our method with the state-of-the-art active learning methods. Furthermore, since random selection is the most straightforward strategy for data selection, we also include it as a baseline in this paper. We first validate our method on classical image classification and semantic segmentation tasks. Then we investigate its performance on simulated cryo-ET data (see Appendix A.4.1), demonstrating its competency for domain tasks. For all the tasks, the labeling budget varies from 10% to 40% corresponding to 7 active learning cycles. In each cycle, 5% of all the training data will be selected for annotation. For the first cycle, we randomly select and annotate 10% of the data from the unlabeled pool and use the selected data as initial training data for all the compared methods. Each reported result is based on three trials to obtain a reliable evaluation. We conduct all the experiments using Pytorch (Paszke et al., 2017) and the source code will be publicly available.

### 4.1 IMAGE CLASSIFICATION

**Datasets.** We adopt four benchmark classification datasets for evaluations, including Cifar10 (Krizhevsky et al., 2009), Cifar100 (Krizhevsky et al., 2009), SVHN (Netzer et al., 2011), and Caltech101 (Fei-Fei et al., 2006). Each Cifar dataset contains 50000 training and 10000 testing samples, uniformly distributed across 10 and 100 classes, respectively. SVHN includes 73257 training and 26032 testing samples, and we do not use the additional training data in SVHN following the standard settings. There are 8677 images of higher resolution (e.g. $300 \times 200$ pixels) in the Caltech101 dataset. We follow the common practice to resize the images to $256 \times 256$, and center-crop the images within a $224 \times 224$ region. Both SVHN and Caltech101 are typical imbalanced datasets. In the latter, for example, the number of samples in a class varies from 40 to 800.

**Model and Training** We use ResNet-18 He et al. (2016) as the task model all the classification experiments. Specifically, we follow the practice in Caramalau et al. (2021); Zhang et al. (2020) to

adopt a special version[3] of ResNet-18 He et al. (2016) as the task model to make the model compatible with the smaller image resolution (i.e. $32 \times 32$) in Cifar10/100 and SVHN. For Caltech101, we use the original ResNet-18. Since an active learning algorithm is generally independent of network architectures, we use the same task model for all the compared methods in each experiment. For instance, on Cifar10, the original CoreSet Sener & Savarese (2018) and VAAL Sinha et al. (2019) employ VGG-16 as the task model which will be replaced by ResNet-18 in our reproduction.

In each cycle of the Cifar and SVHN experiments, we use a batch size of 128 samples to train the task model with an initial learning rate of 0.1. The training continues for 200 epochs and we decay the learning rate with 0.1 at the $160^{th}$ epoch. In Caltech101, the training will continue for 50 epochs and we choose 0.01 as the initial learning rate, which will be decayed by 0.1 at the $40^{th}$ epoch. Due to the higher image resolution, a smaller batch size of 64 is used to avoid GPU memory issues. In order to make the evaluations not impacted by different optimization routines, we employ the same optimizer (i.e. SGD (Bottou, 2010)) for all the experiments of image classification. A momentum of 0.9 and a weight decay rate of $5 \times 10^{-4}$ are used with the optimizer.

**Observations.** We compare our method with the state-of-the-art baselines and illustrate the results in Figure 1. The baseline methods include random selection, MC-Dropout (Gal & Ghahramani, 2016), QBC (Kuo et al., 2018), CoreSet (Sener & Savarese, 2018), VAAL (Sinha et al., 2019), LL4AL (Yoo & Kweon, 2019), SRAAL (Zhang et al., 2020) and GCN (Caramalau et al., 2021). For all these algorithms, we use the recommended hyper-parameters as suggested in their original papers. For the proposed noise stability, we use default hyper-parameters in all experiments: $\sigma^2 = 1$, $K = 10$ and $\xi = 10^{-3} \frac{\|\theta\|}{\|\Delta\theta\|}$ to ensure the relative noise magnitude constrained.

As can be seen in Figure 1, the proposed method (*NoiseStability*) consistently outperforms the baselines (in terms of accuracy) in each active learning cycle. In particular, for harder datasets such as Cifar10, Cifar100 and Caltech, our method achieves remarkably higher final accuracy against the others. For the easier task SVHN, our method yields a more rapid and stable increase of accuracy at the first several cycles, more competitive than the baselines. In addition to the general comparison, we present a detailed analysis below to demonstrate more merits of our method.

- In the later cycles (e.g. 35% and 40%), our method usually exceeds the others by a larger margin (e.g. in Cifar10 and Caltech101). This indicates that when the task model remembers more samples (i.e. the trained model is more likely to behave over-confident), estimating uncertainty by our noise stability is much more accurate than other competitors.

- The superior performance in the Caltech101 experiment suggests that our method is more suitable for imbalanced datasets than the existing baselines. We conjecture this is due to the simplicity of our method, since training helper models (Kuo et al., 2018; Sinha et al., 2019; Zhang et al., 2020; Caramalau et al., 2021) with imbalanced data could be adverse to these helpers themselves, thus leading to an inferior performance of the task model.

- Lastly, we observe that in the Cifar10 experiment, the performance of our method (94.25%) exceeds that (93.6%) of the same model which is trained on the entire dataset without considering active learning. This counter-intuitive observation indicates that training only with a subset of *hard* samples may make DNNs generalize better than training with an entire set.

## 4.2 SEMANTIC SEGMENTATION

Semantic segmentation is more complex than image classification due to the requirement of high classification accuracy at pixel level. As a consequence, in several baseline methods, such as VAAL Sinha et al. (2019), SRAAL Zhang et al. (2020) and GCN Caramalau et al. (2021), the auxiliary models need careful design in order to achieve an satisfactory performance. This increases the burden of deploying AI algorithms in practice. In contrast, our method is free of extra design and can be easily adopted for this problem, demonstrating its task-agnostic advantage.

**Dataset.** We use the Cityscapes Cordts et al. (2016), a large-scale street scene dataset. It includes images taken under various weather conditions in different seasons from 50 cities, making the task on it more challenging. We consider the standard training and validation data of this set. We crop

---

[3]https://github.com/kuangliu/pytorch-cifar

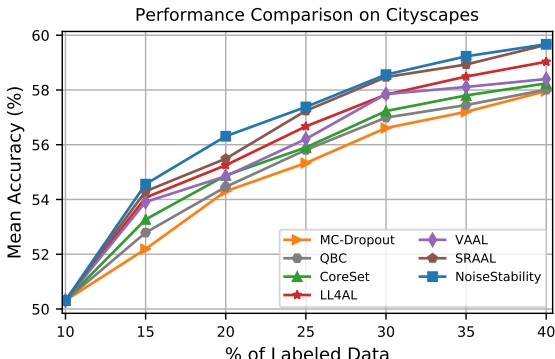

Figure 2: Classification performance due to different active learning methods on the semantic segmentation task Cityscapes.

the images to a dimension of $688 \times 688$, and there are 19 categories (classes) in total for each pixel. It is worth noting that the cropping size also depends on the task model, discussed in next section.

**Model and Training.** As a common practice Zhang et al. (2020), we choose a dilated residual network (DRN-D-22) (Yu et al., 2017) as the task model. It includes 8 layer modules and each module is composed of 1 or 2 *BasicBlock* of layers. One can refer to the original paper (Yu et al., 2017) for the network details .

Similar to the classification evaluations, we train the model for 7 cycles and 50 epochs for each cycle. Following the suggestion in (Yu et al., 2017), we use the Adam optimizer (Kingma & Ba, 2014). We choose a learning rate of $1 \times 10^{-4}$, remained unchanged in the entire training. For the proposed noise stability, we set $K = 3$ and keep the remaining hyper-parameters the same as specified in the classification experiments.

**Observations.** We compare our method with MC-Dropout (Gal & Ghahramani, 2016), QBC (Kuo et al., 2018), CoreSet (Sener & Savarese, 2018), VAAL (Sinha et al., 2019), SRAAL (Zhang et al., 2020) in this experiment. We illustrate the results in Figure 2. We observe that the improvement of the proposed method is solid. Given Cityscapes is highy imbalanced, this experiment once again demonstrates the superiority of our method on imbalanced datasets. More importantly, from classification to segmentation, our method can easily adapt to the new task by simply sampling and adding noise to the output (without being affected by the change of the output format).

## 5 DISCUSSIONS

In this section, we provide further analysis on the effect of the proposed method. First in Section 5.1, we discuss the choice of hyper-parameters used in noise stability. Then we evaluate our algorithm with fewer cycles and larger selection size in Section 5.2. Moreover, in Appendix A.7, we provide an empirical validation for the theoretical hypothesis that selecting samples by noise stability is conducive to reducing the output variance for the training set.

### 5.1 SENSITIVITY TO HYPER-PARAMETERS

We first investigate how our algorithm is sensitive to the hyper-parameters (i.e. the noise magnitude, the noise distribution and the sampling times). We conduct experiments on Cifar10 and Cifar100 for this analysis.

**Noise magnitude.** The criterion of noise stability in Eq (1) is sensitive to the noise magnitude. Specifically, noise with too large magnitudes tends to destroy the learned knowledge in DNNs. For simplicity, we denote by $\lambda$ the relative (to the clean parameter) magnitude, i.e. $\xi = \lambda \frac{\|\boldsymbol{\theta}\|}{\|\Delta \boldsymbol{\theta}^{(i)}\|}$. We evaluate the choices of $\{10^{-5}, 10^{-4}, 10^{-3}, 10^{-2}, 10^{-1}, 1\}$ for $\lambda$. Results in Figure 3 (left) show that

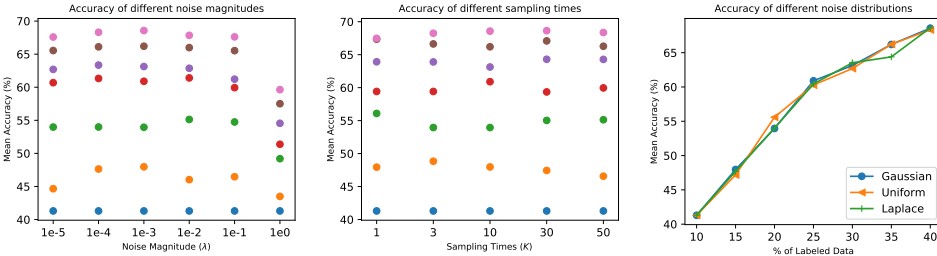

Figure 3: Evaluations of different hyper-parameters on Cifar100: noise magnitude $\lambda$ (**left**), sampling times $K$ (**middle**) and noise distribution (**right**). In the left and middle plots, we use dots with different colors to denote different cycles, e.g. the blue dots at the bottom refer to cycle 1 and the pink dots at the top refer to cycle 7.

using $\lambda$ between $10^{-4}$ and $10^{-2}$ is likely to yield good performance. From the various experiments of this work, we observe that a default choice of $10^{-3}$ would be suitable in general.

**Sampling times.** Accurate approximations such as that for Eq (1) usually require a considerable sampling times. However, we surprisingly find that the noise stability scheme used in our algorithm is very insensitive to the sampling times $K$. We evaluate $K$ with choices of $\{1, 3, 10, 30, 50\}$ and observe very similar results, as shown in Figure 3 (middle). We conjecture this might be because the uncertainty estimation in active learning does not require an accurate approximation of the exact value, even nor the exact order. The main factor that impacts the final results is only the selected subset of data.

**Noise distribution.** Note that our method is not limited to the multivariate Gaussian distribution. Any distribution with all the elements independent of each other satisfies the theoretical properties. Here we evaluate two alternative noise distributions, which are the Uniform distribution $\Delta\boldsymbol{\theta}_i \sim \mathrm{U}(-1, 1)$ and the zero-mean Laplace distribution $\Delta\boldsymbol{\theta}_i \sim \mathrm{Laplace}(0, b)$ with $b = 1$. Results in Figure 3 (right) show that both of them achieve similar good performance as the normal distribution.

## 5.2 Performance on Larger Selection Size

It remains a long-term open problem that whether data uncertainty and diversity are empirically consistent or in a trade-off relationship. Instead of directly addressing this dilemma, we design an experiment to investigate how our algorithm performs with an increased selection size[4]. As the training pool is enlarged by adding a large amount of samples, both uncertainty and diversity tend to be crucial to further training, e.g. highly imbalanced (lack of diversity) data is definitely harmful to learning. For efficiency, we perform a single cycle of sample selection after cycle 1. We use different algorithms to select unlabeled samples with the labeling budgets of 10%, 15%, 20%, 25%, 35% and 40% respectively. Note that it is different from the standard experiments presented in Figure 1, since the sample selection here is one-off rather than iterative. Results in Figure 4 demonstrate that our method still consistently outperforms the SOTA baseline GCN Caramalau et al. (2021).

## 6 Related Work

Active learning has been a long-term open research topic in the community. Most of the methods can be categorized into three types including uncertainty-based, diversity-based and synthesis-based methods. In this section we provide a brief review of these typical studies.

**Uncertainty-based.** Traditional methods solve the problem mainly via estimating uncertainty for unlabeled data. Some typical works (Joshi et al., 2009; Lewis & Catlett, 1994; Lewis & Gale, 1994; Roth & Small, 2006; Tong & Koller, 2001; Vijayanarasimhan & Grauman, 2014; Li & Guo, 2014) have been proposed for shallow or statistical models. Nevertheless, as the rapid development of deep learning, a new challenge has emerged. That is, how to estimate data uncertainty with deep neural

---

[4]Selection size in this section refers to the number of the unlabeled samples selected in a cycle.

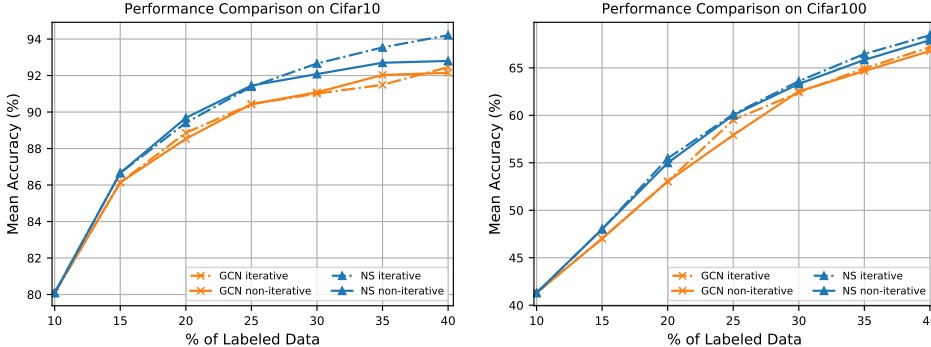

Figure 4: Evaluation of using larger selection size, denoted by *non-iterative* as it only performs an one-off selection. Results with the standard setting (denoted as *iterative*), copied from Figure 1, are also shown for comparison. Experiments are conducted on Cifar10 (**left**) and Cifar100 (**right**).

networks (DNNs). Several methods (Sinha et al., 2019; Yoo & Kweon, 2019; Zhang et al., 2020; Mackowiak et al., 2018) have been proposed to address this challenge with the aid of specialized DNNs and training fashions, such as VAE (variational auto-encoder) (Kingma & Welling, 2013) and adversarial learning. Sener & Savarese (2018); Kuo et al. (2018) attempt to combine a classical algorithm (e.g. K-center or 0-1 Knapsack problem) with DNNs to select the most uncertain data for annotation. Note that our method belongs to this category.

In terms of methodology, the most similar existing work to ours is MC-Dropout (Gal & Ghahramani, 2016). However, the motivation is quite different in that we adopt noise stability to estimate uncertainty, whereas MC-Dropout resorts to Bayesian inference to approximate the uncertainty (measured in the output entropy). Another essential difference lies in that, noise stability only needs a local sampling that generates parameter noise around a specific solution, whereas Bayesian inference requires global sampling over the entire hypothesis space. As a consequence, the necessary sampling times of MC-Dropout (e.g. $\sim 100$ or $\sim 1000$) is orders of magnitude larger than noise stability. Our work is also relevant to recent studies on linearised Laplace approximation of the BNN posterior (Immer et al., 2021; Daxberger et al., 2021). We put a more detailed discussion in Appendix A.8.

**Diversity-based.** In addition to the uncertainty-based methods, several works attempt to select unlabeled data that can diversify the labeled pool. This category of methods are also known as distribution-based (Bilgic & Getoor, 2009; Elhamifar et al., 2013; Freytag et al., 2014; Hasan & Roy-Chowdhury, 2015; Mac Aodha et al., 2014; Nguyen & Smeulders, 2004; Roy & McCallum, 2001; Yang et al., 2015; Settles et al., 2008). Although the diversity-based methods were traditionally treated as essentially different from the uncertainty-based methods, it is worth noting that a recent study (Loquercio et al., 2020) argues that data diversity is correlated to data uncertainty.

**Synthesis-based.** Another typical category of methods aim to diversify the labeled pool by synthesizing new data that can benefit the task model training. Typical methods (Zhu & Bento, 2017; Mahapatra et al., 2018; Mayer & Timofte, 2020) resort to generative models, such as GAN (Goodfellow et al., 2014) or VAE (Kingma & Welling, 2013) to synthesize diverse data samples to train the task model.

## 7 CONCLUSION

This paper proposes a simple method that utilizes noise stability for uncertainty estimation in active learning. For an input sample, noise stability measures how stable can the output be when a random perturbation is added on the model parameters. Our theoretical analysis prove that the noise stability approximates Jacobian norm w.r.t. the parameters assuming the noise magnitude is small. Theoretical and empirical analysis also demonstrate that, selecting those samples mostly unstable to parameter noise has the effect of reducing the prediction variance of the training samples. Experiments on image classification, semantic segmentation, and 3D cryo-ET subtomogram classification demonstrate the efficacy of the proposed method.

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

## A APPENDIX

### A.1 STANDARD MULTIVARIATE GUASSIAN DISTRIBUTION

The univariate Gaussian distribution is often referred to as $\mathcal{N}(\mu, \sigma^2)$, where $\mu$ denotes the expectation of the distribution and $\sigma$ is the standard deviation. $\mathcal{N}(\mu, \sigma^2)$ can be described by its probability density function as $p(x) = \frac{1}{\sqrt{2\pi}\sigma} e^{-\frac{(x-\mu)^2}{2\sigma^2}}$. Multivariate Gaussian distribution $\boldsymbol{x} \sim \mathcal{N}(\boldsymbol{\mu}, \boldsymbol{\Sigma})$, is a generalized form for the vector variable, where $\boldsymbol{\mu}$ is the mean vector and $\boldsymbol{\Sigma}$ is the covariance matrix. In this work, we shall focus on the standard case, in which $\boldsymbol{\mu}$ is the zero vector $\mathbf{0}$ and $\boldsymbol{\Sigma}$ is a diagonal matrix with all diagonal elements being $\sigma^2$, i.e. $\boldsymbol{\Sigma} = \sigma^2 \mathbf{I}$. Note that if $\boldsymbol{x} \sim \mathcal{N}(\mathbf{0}, \sigma^2 \mathbf{I})$, all the samplings $\{\boldsymbol{x}_i\}$ are independent of each other, and each conforms to the standard univariate Gaussian distribution as $x_i \sim \mathcal{N}(0, \sigma^2)$.

### A.2 PROOF OF EQ (5)

We first assume $f(.; \boldsymbol{\theta})$ is a real-valued function and the Jacobian is a vector. If $\Delta\boldsymbol{\theta} \in \mathbb{R}^d$ conforms to the standard multivariate Gaussian distribution that $\Delta\boldsymbol{\theta} \sim \mathcal{N}(\mathbf{0}, \sigma^2 \mathbf{I})$, each of its elements are independent with each other. That is, for any $1 \leq i, j \leq d$, the following is true:

$$\mathop{\mathbb{E}}_{\Delta\boldsymbol{\theta} \sim \mathcal{N}(\mathbf{0}, \sigma^2 \mathbf{I})} \Delta\boldsymbol{\theta}_i \Delta\boldsymbol{\theta}_j = \sigma^2 \delta_{ij}, \tag{11}$$

where $\delta_{ij} = \mathbf{1}_{i=j}$. Without ambiguity, here we use J as a shorthand for $\mathrm{J}_{\boldsymbol{\theta}}(\boldsymbol{x}; \boldsymbol{\theta})$. The notation of Gaussian distribution under the expectation is also omitted for simplicity.

*Proof.* Based on Eq (11), we can expand the inner product between J and $\Delta\boldsymbol{\theta}$ in Eq (4) as

$$
\begin{aligned}
R(\boldsymbol{x}) &= \xi^2 \mathop{\mathbb{E}}_{\Delta\boldsymbol{\theta}} \| \sum_{i=1}^{d} \mathrm{J}_i \Delta\boldsymbol{\theta}_i \|^2 \\
&= \xi^2 \mathop{\mathbb{E}}_{\Delta\boldsymbol{\theta}} \sum_{i=1}^{d} \mathrm{J}_i^2 \Delta\boldsymbol{\theta}_i^2 + \xi^2 \mathop{\mathbb{E}}_{\Delta\boldsymbol{\theta}} \sum_{\substack{i,j=1 \\ i \neq j}}^{d} \mathrm{J}_i \Delta\boldsymbol{\theta}_i \mathrm{J}_j \Delta\boldsymbol{\theta}_j \\
&= \xi^2 \sum_{i=1}^{d} \mathrm{J}_i^2 \mathop{\mathbb{E}}_{\Delta\boldsymbol{\theta}} \Delta\boldsymbol{\theta}_i^2 + \xi^2 \sum_{\substack{i,j=1 \\ i \neq j}}^{d} \mathrm{J}_i \mathrm{J}_j \mathop{\mathbb{E}}_{\Delta\boldsymbol{\theta}} \Delta\boldsymbol{\theta}_i \Delta\boldsymbol{\theta}_j \\
&= \xi^2 \sum_{i=1}^{d} \mathrm{J}_i^2 \sigma^2 \\
&= \xi^2 \sigma^2 \| \mathrm{J} \|^2.
\end{aligned}
\tag{12}
$$

When $f(.; \boldsymbol{\theta})$ is vector-valued, we can straightforwardly put each row of the Jacobian matrix into Eq (12) and finally get

$$R(\boldsymbol{x}) = \xi^2 \sigma^2 \| \mathrm{J} \|_F^2. \tag{13}$$

$\square$

Note that, the conclusion is true for any other zero-mean distribution as long as it satisfies the independence prerequisite and has a definite variance. The multivariate Gaussian distribution is used only as a typical example.

### A.3 MORE COMPARISONS WITH DEEP BAYESIAN ACTIVE LEARNING

In this section, we compare our method with more deep Bayesian active learning methods on MNIST and Cifar10. The deep Bayesian method is one of the most important schemes for uncertainty estimation. We evaluate MC-Dropout Gal & Ghahramani (2016), BALD Houlsby et al. (2011), BatchBALD Kirsch et al. (2019) and Checkpoint Ensemble Chitta et al. (2019). For all the dropout based methods (MC-Dropout, BALD and BatchBALD), which can be regarded as a kind of implicit ensemble, we set the number of Monte-Carlo sampling to 50. For Checkpoint Ensemble, we save the checkpoints for the last 10 epochs as the ensemble models. BALD (mutual information) is used as the acquisition function for the Ensemble method. For our method, we still sample noise for 10 times.

**Model and Training.** For Cifar10, we use ResNet-18 with exactly the same settings as described in Section 4.1. For MNIST, we employ a small convolutional network composed of two 5×5 convolutional layers, followed by two fully connected layers. We use the Adam optimizer with a learning rate of 0.001. For each AL cycle, we train the model for 50 epochs on the labeled pool. The initial labeled pool consists of 50 samples, and another 50 unlabeled samples are selected for annotation at each AL cycle.

**Observations.** We present the results in Figure 5. We observe that the basic deep Bayesian methods, such as MC-Dropout and BALD, perform better than Random selection on both datasets. The advanced Bayesian method BatchBALD does not show superiority, compared with BALD on MNIST. On Cifar10 with a selection size of 2500, BatchBALD almost degenerates to Random selection, and we conjecture this is due to the inaccurate joint entropy estimation. Though using a smaller sampling number, the Ensemble and our NoiseStability methods perform the best. Similar to our method, for MC-Dropout, BALD and BatchBALD, further increasing the number of Monte-Carlo samplings (e.g. an increment to 100) cannot bring any performance gain on the two datasets.

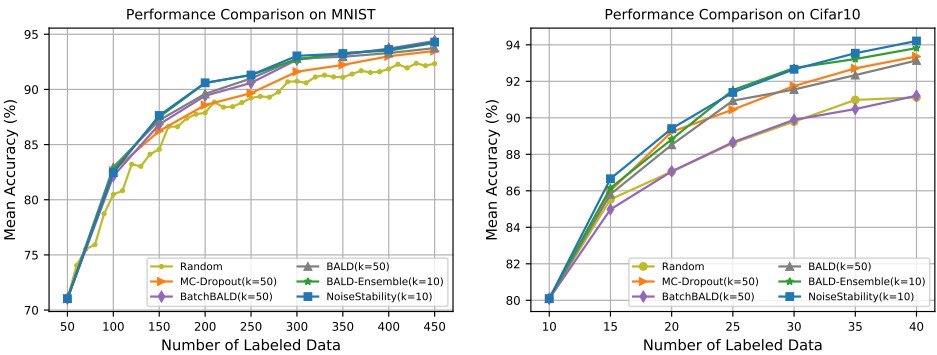

Figure 5: Classification performance due to different Bayesian active learning methods on MNIST (**left**) and Cifar10 (**right**).

### A.4 EXPERIMENTS ON ADDITIONAL SETTINGS

Here we present more experimental results to evaluate our algorithm.

### A.4.1 EXPERIMENTS ON MORE DATA DOMAINS

In this part, we evaluate AL methods on the Cryo-ET Challenges. Cryo-Electron Tomography (cryo-ET) is an important imaging technique to study biological macromolecules and cells, used to produce their high-resolution 3D tomograms of cellular landscapes and structures. Active learning is of great interest for such kind of tasks since the annotation cost is extremely high. We evaluate our active learning algorithm on cryo-ET subtomogram classification (Chen et al., 2017), demonstrating the capability of our method in various domains.

**Dataset.** We compare our noise stability against the baseline methods on one simulated cryo-ET dataset consisting of subtomogram images contaminated by noise with a SNR (signal-noise-ratio)

of 0.03. Given different conformational states, all the samples in this dataset are categorized into 50 classes, namely *50c-snr003*. There are 24000 training and 1000 test samples, uniformly distributed across all the classes. One can refer to Gubins et al. (2019) for how to generate the data.

**Model and Training.** The dimension of the subtomogram data (i.e. $32 \times 32 \times 32$) is different with that of RGB images, making the classification task more challenging. To adapt to this particular data dimension, we employ a customized ResNet-18 with 3D convolutional layers as the task model. In addition to the task model, in the baseline methods one also needs to customize the auxiliary models (e.g. VAE in VAAL (Sinha et al., 2019) and SRAAL (Zhang et al., 2020), and the loss prediction module in LL4AL (Yoo & Kweon, 2019)), whereas our method is free of this extra implementation.

The SGD optimizer (Bottou, 2010) is employed to train the task model. The training will continue for 100 epochs and the initial learning rate is set as 0.1, and we decay it by 0.1 at the $80^{th}$ epoch. We choose a momentum of 0.9 and a weight decay rate of $5 \times 10^{-4}$, respectively. For the proposed noise stability, we set $K = 5$ and remain the hyper-parameters the same as specified in the classification experiments.

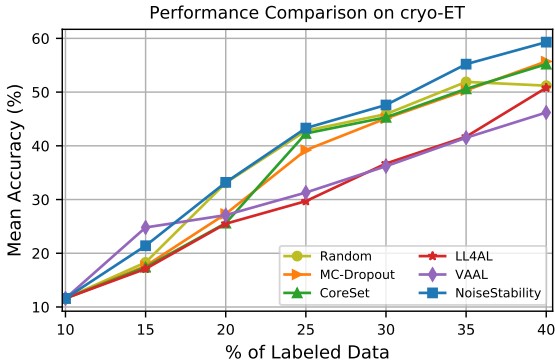

Figure 6: Classification performance due to different active learning methods on the cryo-ET task: 50c-snr003.

**Observations.** We present the comparisons of the different methods in Figure 6. At each cycle, our method demonstrates solid improvement over the others. Moreover, in the later cycles, our method outperforms the others by a significant margin, demonstrating its consistent superiority. Moreover, due to the noise in the data, some other methods show reduced accuracy as cycle increases (e.g. CoreSet (Sener & Savarese, 2018), whereas our method keeps climbing towards a higher accuracy.

### A.4.2 EXPERIMENTS ON REGRESSION PROBLEMS

Here we show that our method can also be applied to regression problems. We evaluate several active learning algorithms on Ames Housing dataset De Cock (2011) to predict house prices.

**Dataset.** The Ames Housing dataset De Cock (2011) describes the sales of individual residential properties in Ames, Iowa from 2006 to 2010. The data set includes 1461 labeled training examples. There are 60 explanatory variables involved in assessing home values. The Ames Housing dataset is considered an expanded version of the classical Boston Housing dataset, which has only 506 labeled samples and 14 attributes.

**Model and Training.** We use 50% of the data for training set and the remaining for testing. We follow common practices to transform the categorical features into numerical ones with one-hot encoding. A four-layer MLP is employed to extract features of the input vectors, followed by another fully connected layer as the predictor. To adapt the regression task, we re-implement MC-Dropout to compute the output variance among multiple predictions. Methods such as BALD Houlsby et al. (2011) and BatchBALD Kirsch et al. (2019), which rely on the output entropy are not suitable for regression tasks.

The Adam optimizer is employed to train the task model. The training will continue for 300 epochs and the learning rate is set as 0.001. For the proposed noise stability, we set $K = 10$ and remain the

hyper-parameters the same as specified in the classification experiments. For MC-Dropout, we set the number of Monte-Carlo sampling to 50 as usually recommended.

**Observations.** We start the training with 20 labeled samples and iteratively acquire 20 unlabeled samples for annotation. As illustrated in Figure 7, our NoiseStability yields the lowest mean absolute error in most places. Moreover, our selected data in earlier stages reduces the error more quickly than the others, demonstrating the advantage of our method when there is less labeled data in training. As the training gets into later stages, our method also shows a superiority than most of the others, indicating its stability when more labeled data is involved in training.

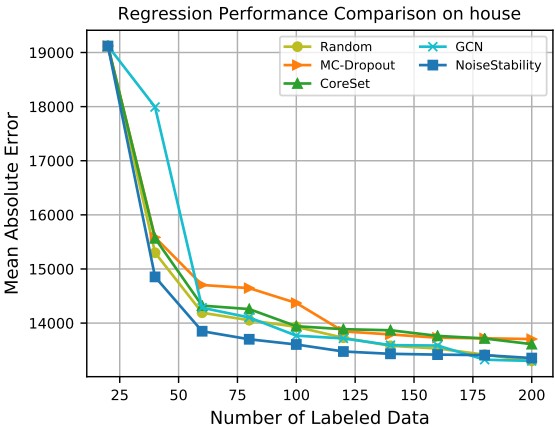

Figure 7: Regression performance due to different active learning methods on the Ames Housing dataset. The mean absolute error is compared. Lower is better.

### A.4.3 EXPERIMENTS WITH SMALL SELECTION SIZE

To further validate the effectiveness of our method, we consider smaller selection size 10 and 1 respectively on MNIST. Though such very small selection sizes are not frequently used in practice, they can be treated as a popular indicator to evaluate the performance of an acquisition function. Moreover, under the extreme case in which the selection size is 1, results can reveal whether an AL algorithm will successfully select the most informative or representative samples at each AL cycle, without concerning the problem of batch acquisition.

We use the same settings as in the MNIST experiments described in Appendix A.3. As demonstrated in Figure 8, all the algorithms perform consistently when different selection sizes are used, i.e. algorithms perform well with larger selection sizes also have good performance with smaller selection sizes. Under smaller selection sizes, all the state-of-the-art methods outperform Random selection by a large margin. Moreover, our NoiseStability method yields the highest accuracy.

### A.4.4 EXPERIMENTS ON LARGE SCALE DATASETS

Our method can be easily applied to large scale datasets with a large selection size. Here, we compare our method with Random selection and MC-Dropout on such a large dataset, namely ImageNet Deng et al. (2009).

**Dataset.** The ImageNet dataset Deng et al. (2009) consists of a total of 14 million images with more than 20K categories. We use the popular subset ILSVRC2012 which includes around 1.3M images distributed across 1K categories. We randomly select 15% of the total training samples to build the initial labeled pool and iteratively select 5% of the total training samples for annotation. Note that the data selection is conducted without replacement, as we do in all the other experiments.

**Model and Training.** We employ the standard ResNet-18 model for this task. SGD is used as the optimizer with an initial learning rate of 0.1, momentum of 0.9, weight decay of 0.0005 and batch size of 256. We train the model for 90 epochs following the practice in Pytorch official code. The learning rate decays by a factor of 0.1 every 30 epochs.

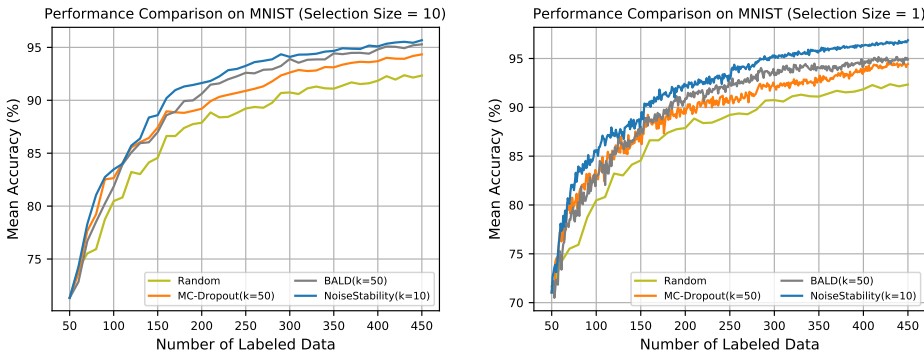

Figure 8: Classification performance due to different Bayesian active learning methods on MNIST with a selection size of 10 (**left**) and 1 (**right**), respectively.

**Observations.** As illustrated in Figure 9, MC-Dropout and Noise Stability are consistently superior to the naive Random selection. Our method significantly outperforms the two baselines at the later AL cycles.

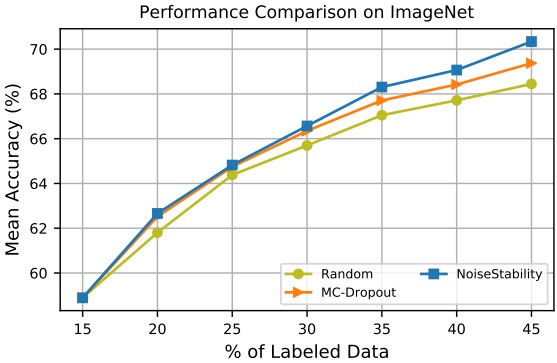

Figure 9: Classification performance due to different active learning methods on the ImageNet dataset. Top-1 Accuracy is reported.

### A.4.5 EXPERIMENTS ON MORE ARCHITECTURES

Since the proposed method directly manipulates the task model, we conduct additional experiments on another architecture in order to validate the architecture-agnostic nature of the proposed method. Specifically, we choose MobileNet-v2, a compact yet effective DNN architecture with fewer parameters. We evaluate the AL algorithms on Cifar10 and Cifar100, using exactly the same settings as in the experiments in which ResNet-18 is used. As shown in Figure 10, our method still yields very competitive performance. In Cifar100, our method is on par with CoreSet, but it outperforms the others by a decent margin. Note that in the regression experiment in Appendix A.4.2, our method is also shown effective on MLP architectures. These results demonstrate that our method is independent of model architectures.

### A.5 ADDITIONAL ABLATION STUDY

Here we conduct an ablation study to analyze the two important designs in our method. One is the adaptive noise magnitude. The other is the position to inject noise.

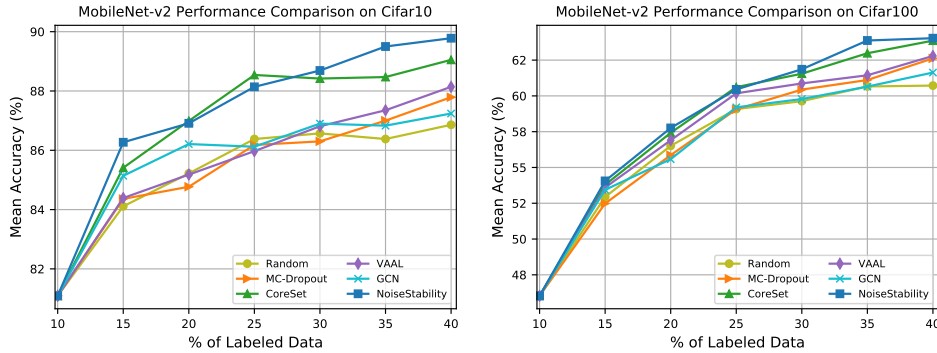

Figure 10: Classification performance due to different active learning methods on Cifar10 (**left**) and Cifar100 (**right**), using MobileNet-v2 as the backbone.

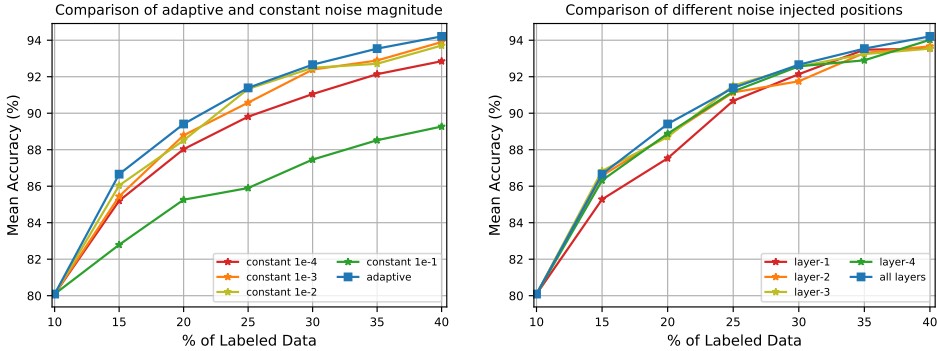

Figure 11: Ablation study of our NoiseStability method on Cifar10. We present the effect of adaptive noise magnitude (**left**) and the position to inject noise (**right**).

### A.5.1 ADAPTIVE NOISE MAGNITUDE

The noise magnitude should be proper, i.e. a very large magnitude will destroy the network and a very small one makes the model less capable to distinguish uncertain samples. We adopt an adaptive strategy to control the noise magnitude. In order to validate its effectiveness, we evaluate our method with several constant noise magnitudes including 1e-1, 1e-2, 1e-3 and 1e-4. As shown in Figure 11 (**left**), our method performs much worse if the constant magnitude is too small or too large. When a proper constant is used, i.e. 1e-2 or 1e-3, the performance is comparable to the adaptive strategy. However, selecting a reasonable constant noise magnitude for each specific model sounds ad-hoc, since model parameters w.r.t. different architectures usually have different value distributions. More importantly, our experimental results on ResNet-18, MobileNet-v2 and MLP have demonstrated that the adaptive noise magnitude works well across various architectures, demonstrating the rationale of using the adaptive strategy.

### A.5.2 POSITION OF APPLYING NOISE

Here we investigate how the position of applying noise affects the performance. We take ResNet-18 as the task model that includes four *layer modules*, each of which is composed of a sequence of *BasicBlocks*. In this experiment, we inject noise to each *layer module* respectively. As shown in Figure 11 (**right**), changing the position of noise injection does not affect the performance much.

### A.6 RUNNING TIME COMPARISON

To demonstrate the time efficiency, we report the running time of the acquisition function used in each AL baseline in Table 1. We conduct the experiments on Cifar10, with a selection size

of 2500 and 250, respectively. The size of the corresponding unlabelled pool is 10000 and 20000, respectively. The running time is measured on a single Nvidia Tesla V100 GPU. For fair comparison, we set the number of dropout sampling to 10 for all the Bayesian methods. As can be seen, our NoiseStability is relatively efficient among these AL baselines. Moreover, the time complexity of our method increases approximately linearly as the size of the unlabelled pool is increasing, but is almost independent of the selection size.

| **Method** | Unlabeled Pool 10000 | | Unlabeled Pool 20000 | |
|---|---|---|---|---|
| | 250 Selected | 2500 Selected | 250 Selected | 2500 Selected |
| MC-Dropout | 1.01 | 1.02 | 1.92 | 2.25 |
| CoreSet | 2.19 | 49.26 | 6.15 | 54.72 |
| LL4AL | 0.13 | 0.28 | 0.30 | 0.48 |
| VAAL | 225.96 | 402.32 | 270.18 | 462.24 |
| GCN | 0.36 | 1.10 | 0.99 | 1.34 |
| BALD | 1.53 | 2.10 | 3.39 | 4.08 |
| BatchBALD | 7.29 | 77.17 | 18.17 | 115.66 |
| NoiseStability | 2.23 | 2.28 | 4.81 | 5.04 |

Table 1: Data selection time (in minutes) for the first cycle on Cifar10 with two selection sizes 2500 and 250. All the results are measured on a Nvidia Tesla V100 GPU with an Intel(R) Xeon(R) Gold 5117 CPU.

## A.7 EMPIRICAL VALIDATION OF VARIANCE REDUCTION

The theoretical analysis in Section 3.2 reveals the connection between our NoiseStability and variance reduction. Here we validate this connection, by quantifying the variance reduction for the training samples used in cycle 1, using the model trained in the subsequent cycle. As the original form of output variance is intractable for DNNs (Thisted, 1988), we choose MC-Dropout (Gal & Ghahramani, 2016) as the criterion for variance estimation to evaluate our method and the others. Specifically, we approximate the uncertainty with 1) the confidence represented by Bayesian posterior probability for the true label and 2) the prediction entropy. As illustrated in Table 2, while training with more samples always reduce the output variance, selecting samples with our NoiseStability exhibits the best effect.

Table 2: Evaluation of variance reduction for different active learning methods at cycle 2 on Cifar10. The confidence of the true label (higher is better) and Entropy (lower is better) are used to approximate the variance. We use the same random seed in the all experiments. The Confidence and Entropy at cycle 1 for training samples are 0.64 and 0.69, respectively.

| Criterion | Random | CoreSet | VAAL | GCN | NoiseStability |
|---|---|---|---|---|---|
| Confidence | 0.77 | 0.79 | 0.74 | 0.78 | **0.80** |
| Entropy | 0.45 | 0.44 | 0.51 | 0.44 | **0.42** |

## A.8 DISCUSSION ABOUT MORE RELATED WORKS

Studies in active learning with a linearised Laplace approximation of the BNN posterior (Immer et al., 2021; Daxberger et al., 2021) are very relevant to our method. Here we briefly discussing their connections and differences. From a high level, our criterion is the minimum mean-squared error estimation (MMSE) while theirs are Bayesian estimation, corresponding to the frequentists' and Bayesian perspective in statistics. They are not incompatible, yet they are inextricably linked. For instance, under the jointly Gaussian conditions (such as the Gaussian priors assumed in those papers) the MAP and MMSE are known to be equivalent; They face similar computational challenges as MMSE is the conditional mean that are as difficult to calculate as the MAP. Our approach is to approximate full conditional mean by the linear MMSE, which only pertains to the mean and covariances. This is a principled approach that is also used in practice in other areas such as the Kalman filter. In contrast, the Laplace approximation (e.g. (3) in Immer et al. (2021)) is accurate only when $\theta$ is close to the optimal value $\theta^*$ (where the first order term is negligible) and thus the

theory only provides local guarantees. It is our perspective that the condition is restrictive and does not apply to practical deep learning methods. The resulting algorithms are also different: Immer et al. (2021) is based on the determinant of the Hessian matrix; the method in Daxberger et al. (2021) involves the diagonal of inverse Hessian. According to our findings, the predictive variance reduction conditioning on any direction is always monotonic decreasing with the gradient norm and servers as the cornerstone of our approach.

