# OpenReview forum: "Deep Active Learning with Noise Stability"
_ICLR.cc/2022/Conference — ICLR 2022 Submitted_

### Official Review · Reviewer_9WDB · 2021-11-02

**Correctness:** 3
**Technical Novelty And Significance:** 3
**Empirical Novelty And Significance:** 3
**Recommendation:** 6
**Confidence:** 4

**Main Review:**

+:

+ The method is simple and does not involve auxiliary models.
+ Experiments are performed on several tasks and datasets
+ The paper provides some theoretical connection to variance reduction (they are straightforward though).


-:
- The clarity of the presentation can be improved in several places. For example:
It seems that in the current presentation, the assumption is that epsilon (noise magnitude) is the same for all parameters. It is not clear if this is implemented. Also, is this a valid assumption in terms of noise being relatively “small"? Is the noise added to all parameters? One or more baselines are missing from some of the results figures without any explanation.

- Since the proposed method directly works on the task model as opposed to some of the baselines, it would be helpful to see more models/architectures tested for each task.

- The proposed method does not seem specific to image data, so evaluating it on tasks in other domains would be beneficial to validate its applicability across different domains.


**Summary Of The Paper:**

A method for batch active learning for DNNs is proposed in the paper. The main idea is to consider robustness to parameter perturbation as an uncertainty measure. In other words, prediction deviation when adding a small noise to the parameters is used as a score for selecting data to be labeled in the active learning loop. The authors provide some theoretical analysis for the method to connect it to variance reduction. Experiments on several image classification and semantic segmentation datasets show that the method performs better than several SotA baselines.

**Summary Of The Review:**

Overall, the paper proposes a simple method for active learning, provides some theoretical connection with existing literature, and shows its effectiveness where it improves SotA. While there is room for improving the paper, I tend towards acceptance.

--Update--
I think the authors have addressed most of the concerns and comments, and I remain positive about the paper.

---

> ### Author Response · Authors · 2021-11-23
> **Thank you for your valuable comments**
>
> We thank you for your constructive and valuable comments. We provide clarifications below and we hope they can address your concerns. But please also feel free to raise any questions during the discussion phase and we will be happy to answer the questions.
>
>
>
> **1. The clarity of the presentation can be improved in several places. For example: It seems that in the current presentation, the assumption is that epsilon (noise magnitude) is the same for all parameters. It is not clear if this is implemented.**
>
> We are sorry for the misleading. In fact, we do not set the magnitude for “*each parameter*”. In this paper, we take the parameters $\mathbf{\theta}$ (and also the injected noise $\Delta \mathbf{\theta}$) as a high-dimensional vector, and multiply the entire vector $\Delta \mathbf{\theta}$ by the scalar $\xi$. We added a brief clarification at the end of Section 2.2.
>
> **2. Also, is this a valid assumption in terms of noise being relatively “small"?**
>
> Injected noise with a relative magnitude of 0.001 can be regarded as small enough as suggested by previous studies such as "*Stronger generalization bounds for deep neural networks via a compression approach*" by Arora, et.al. Empirically, we find that the output deviations under this magnitude are reasonable. Additional ablation study in Appendix A.5.1 shows that, when the noise magnitude is not small enough, uncertainty estimation with noise stability suffers drastic performance drop.
>
>
> **3. Is the noise added to all parameters?**
>
> Yes, the noise is added to all parameters except for the scaling and shifting parameters used in Batch Normalization (BN) layers.
>
>
> **4. One or more baselines are missing from some of the results figures without any explanation.**
>
> GCN results are not included in SVHN, Caltech and Cityscapes datasets since their released code causes a memory access error on these datasets. For cryo-ET, we only compare some selected baselines due to high compute costs. In the revised manuscript, we added Random and MC-Dropout in the cryo-ET experiment in Figure 6. We find that some state-of-the-art methods (e.g. LL4AL and VAAL) show worse performance than Random selection, implying that these methods may not be suitable for this domain or need more careful domain-specific designing. Other missing baselines will be added in the next version.
>
>
> **5. Since the proposed method directly works on the task model as opposed to some of the baselines, it would be helpful to see more models/architectures tested for each task.**
>
> We conducted additional experiments on MobileNet-v2, which is an effective and compact architecture. Results in Appendix A.4.5 demonstrate the superior performance of our method on different model architectures.
>
>
> **6. The proposed method does not seem specific to image data, so evaluating it on tasks in other domains would be beneficial to validate its applicability across different domains.**
>
> We have evaluated our method on a cryo-ET dataset (Now moved to Appendix A.4.1), which is not a typical computer vision task. The input of cryo-ET is not an RGB image. In the revised manuscript, we also presented evaluations on a regression task with numerical features as input. For details, please refer to Appendix A.4.2, in which we show that our method outperforms the SOTA baselines.

---

### Official Review · Reviewer_pooj · 2021-11-03

**Correctness:** 1
**Technical Novelty And Significance:** 2
**Empirical Novelty And Significance:** 2
**Recommendation:** 3
**Confidence:** 5

**Main Review:**

Strengths:
- The idea is simple and easy to implement.
- Results seem to generalize to different datasets.

Weaknesses:
- Links to calibration are confusing and not empirically supported.
- Results are not convincing:
a) Large scale is needed (Imagenet for conclusive results)
b) Ensembles are not compared (and MC Dropout results are somehow questionable). See below
c) Compute cost is not provided.

**Summary Of The Paper:**

This paper proposes a single-model active learning method for classification and segmentation. The key idea is to add noise to the parameters in order to generate multiple outputs and be able to estimate the uncertainty of a sample. The paper presents results for classification and segmentation in different datasets.

**Summary Of The Review:**

The idea behind the paper is interesting. Instead of using MC Dropout of Ensemble methods trained with different seeds, use a single model and do multiple forward passes adding a perturbation to the parameters for computing the output distribution needed to estimate uncertainty.

As this is basically a multi-model approach (using variations of a single model), would be interesting to see a comparison with ensembles. In the end, ensembles are the best approximation to Bayesian Uncertainty estimation.
For MC Dropout, I wonder about the parameters. I do believe the results are taken from one of the references, but details there are needed. MC Dropout easily outperforms single models like Entropy when the number of forward passes is large enough. How many models are used here? Would be good to see, at least 50-100 forward passes. It is actually surprising that MC Dropout performs on par or worse than random on CIFAR.

Same with the proposed method. How does the method vary as a function of the number of forward passes? How does that compare to the same ensembles? A similar comparison was done (for classification) in: Chitta et al. Training Data Subset Search with Ensemble Active Learning arXiv 2020. The number of forward passes needed is larger compared to models trained with different seeds.

The paper talks about uncertainty calibration but there is no empirical evidence. That would be very interesting to see.

I would also be interested in understanding how the noise is applied to the layers and if noise in all layers is needed. That would have an impact in the inference time needed for computing uncertainty. An ablation study here would be interesting.

Same for cityscapes. I am missing Random as comparison.


For all the experiments, I would suggest adding Entropy as a baseline for comparison. Entropy is straightforward and provides very solid results.

---

> ### Author Response · Authors · 2021-11-23
> **Thank you for your valuable comments**
>
> We thank you for taking the time to review the paper and give valuable comments. Below we summarize and address main concerns mentioned in **weaknesses** and **summary of the review**. Please also feel free to raise any questions during the discussion phase and we will be happy to answer the questions.
>
>
> **1. Links to calibration are confusing and not empirically supported.**
>
> Since we did not claim any link to calibration, we wonder whether the reviewer means the link to variance reduction? We have provided an empirical validation for variance reduction in the original version (now moved to Appendix A.7). We compute the change of variance, measured by the predictive confidence and entropy w.r.t. the Bayesian posterior probability. Results show that our method indeed reduces the variance as expectation.
>
>
> **2. Results are not convincing: a) Large scale is needed (ImageNet for conclusive results)**
>
> We added experiments on ImageNet. Due to the resource and time limitation, we only compare our method with *Random* and *MC-Dropout* (50 feed-forward passes for Monte-Carlo sampling). Our method is superior to these two baselines.
>
>
> **3. Results are not convincing:  b) Ensembles are not compared**
>
> We added experiments on MNIST and CIFAR10 to compare with the ensemble methods in Appendix A.3. As the training cost of ensemble methods is extremely high, we adopt the algorithm proposed in Chitta et al. "Training Data Subset Search with Ensemble Active Learning arXiv 2020." We use checkpoints of the last 10 epochs as ensemble models. Results show that our method is slightly superior. We conjecture the reason might be as follows: though ensemble provides more accurate Bayesian Uncertainty estimation (than MC-Dropout), a really reliable Bayesian inference for deep models is still intractable due to the extremely large hypothesis space. Our method has connections with the Jacobian norm of the target sample, which may provide useful clues to distinguish informative samples.
>
>
> **4. Results are not convincing: b) MC Dropout results are somehow questionable**
>
> We are sorry that the results of MC-Dropout in the previous version were indeed not reliable as we only added dropout between two convolutional layers in the BasicBlock of ResNet-18. After careful investigation, we find that adding dropout after each convolutional layer (as suggested in BatchBALD) achieves competitive results. We update all results of MC-Dropout, including Cifar10, Cifar100, SVHN and Caltech. For those newly added experiments, we use the correct implementation of MC-Dropout. In all these experiments, we set the number of feed-forward passes to 50 during data selection.
>
>
> **5. Results are not convincing: c) Compute cost is not provided.**
>
> We provided the runtime comparison in Appendix A.6. Our method is relatively efficient among various AL baselines.
>
>
> **6. How does the method vary as a function of the number of forward passes.**
>
> As shown in Figure 3 (middle plot), our method is insensitive to the number of noise samplings. So we set the sampling number to 10 (or even fewer) for all of the experiments for our method. Using the correct implementation of MC-Dropout, we compare our method with  the sampling number of 10 and MC-Dropout with the sampling number of 50 in all experiments. Results in Figure 1 and Figure 5 demonstrate that our method consistently outperforms MC-Dropout. We also find that, similar as our method, further increasing the sampling number (e.g. an increment to 100) cannot bring any performance gain for MC-Dropout in the MNIST and Cifar10 experiments.
>
>
> **7. How the noise is applied to the layers and if noise in all layers is needed.**
>
> We apply the noise by directly adding it to the model parameters of the layers. We conduct an ablation study in Appendix A.5.2 to show that applying the noise to all the layers achieves better performance than only adding the noise to several layers.
>
>
> **8. For all the experiments, I would suggest adding Entropy as a baseline for comparison**
>
> Due to resource limitation, we did not conduct experiments with Entropy, since several baseline methods (e.g. MC-Dropout, BALD) have already been demonstrated superior than the Entropy method. However, we agree that Entropy should be added as a baseline for completeness and will add it in a future version, as well as Random selection in Cityscapes.

---

> > ### Comment · Reviewer_pooj · 2021-12-08
> > **Results concerns still apply**
> >
> > I appreciate the authors' efforts in answering my questions and adding new numbers.
> >
> > However, the concerns are still there, so I will keep my score.
> >
> > - There are partial comparisons on Imagenet. I understand there was not enough time, but that should not be an excuse. I would encourage completing these results for a stronger submission.
> >
> > - I am surprised there are no comparisons with Entropy. That is a basic single model method that is, actually, quite effective even compared to other approaches. I understand the authors' comment that those values are not included in other publications, but in my experience, Entropy is much better than some existing, publishing methods. I do encourage adding those numbers for comprehensiveness.
> >
> > - Last but not least, the authors reveal the issue with the MCDropout numbers in the paper. Adding a 50 passes dropout reduces the performance gap which is discouraging. Usually, MCDropout should take closer to 100 passes rather than 50. Again, I encourage the authors to add proper comparisons here. In the current from seems like selected results.
> >
> > - I am also surprised the method is insensitive to the number of forward passes. At some point there should be an inflection point, maybe 2 passes? and at some point there should be a benefit of doing many more (as in MCDropout)
> >
> > - On the positive side, as mentioned before, the benefit of this approach, if proven successful, is how easy it would be to apply to different tasks

---

### Official Review · Reviewer_owNu · 2021-11-03

**Correctness:** 3
**Technical Novelty And Significance:** 3
**Empirical Novelty And Significance:** Not applicable
**Recommendation:** 5
**Confidence:** 3

**Main Review:**

## Strengths

* **Relevance**: The active learning problem is an important one in today's age of 'big data' and deep learning. Deep neural networks are particularly data-hungry and we wish to be able to train them with as few data points as possible. However, as noted by the authors, deep neural networks are also pathologically overconfident in their predictions. Thus, predictive entropy (for example) cannot be reliably used as a proxy for the informativeness of unlabelled examples. Therefore, methods for AL with deep neural networks are of high relevance and importance to the field.

* **Simplicity**: The proposed method is simple to understand and implement. Many state-of-the-art AL methods rely on complicated Bayesian approximations and are difficult to understand/implement. A simple method that works well is a good method!

## Weaknesses

* **Clarity**: I found the paper hard to understand in many places and there were many grammatical errors. Overall, I think a lot more effort needs to be put into making the ideas presented easy to understand and follow. I'll provide some specific examples here, however, this is a general comment for the paper. I encourage the authors to spend more time thinking about how best to present their ideas. I also encourage them to use a tool like Grammarly which will flag problematic sentences.
    1.  "This is also aligned with the pre-requisite of semi-supervised learning." – What is the pre-requisite of semi-supervised learning? What is the point of this sentence?
    2. "... namely uncertainty based method, aims to select a portion of the most uncertain or representative data ..." – uncertain is not the same as representative so it is not clear exactly what is meant by this sentence.
    3. What is the "interlayer cushion"? This seems like a pretty specific term so it should be explained.
    4. What does "tends to be easy to recognize future examples" mean? Is this to say that the model is good at determining which examples should be labelled?
   5. "Specifically, we introduce a simple algorithm of uncertainty estimation by computing the distance between outputs of the clear and perturbed **input** ..." (emphasis my own) – this sentence makes it sound like the inputs are also perturbed, when in fact it is only the model parameters that are perturbed.
    6. "Our method is easy to implement and free of customized auxiliary models. Therefore, ..." – I think a bit more context is needed for what kinds of auxiliary models other AL methods require, in order for this statement to make sense to the reader.
    7. "pool of unlabelled data" – I believe the standard term is "pool set".
    8. "labelled pool" – this is just the train set.
    9. I don't think the set update equations are needed to understand the moving of labelled examples from the pool set to the train set. If anything the equations make this seem more complicated than it actually is.
    10. In section 3.2 the authors switch notation for the Jacobian of the output w.r.t the parameters. This doesn't seem necessary to me and makes things more difficult to follow.
    11. "Note we omit the notation of \theta in J and A to avoid ambiguity" – presumably the authors mean that the omission is okay because there is no ambiguity?
    12. Where does equation 7 come from? Some elaboration would be helpful.
    13. In equation 10, in the last line the numerator contains a J_\hat{y}(x)^T. Should this be u(x)?


* **Experimental evaluation**: while the experimental evaluation provides a good start (the ablations provided are especially appreciated), it is not of the standard required for an ICLR paper, in my opinion. I have concerns with:
    1. **The narrow range of underlying tasks.** The tasks considered in the paper are all image classification tasks of various kinds. Regression tasks and non-computer vision tasks should be added.
    2. **Missing baselines.** In particular, batch-aware acquisition strategies such as BatchBALD (Kirsch et al.) and sparse subset approximation (Pinsler et al.). The BALD acquisition function  (Houlsby et al.) is also missing and would likely improve the strength of the Dropout baseline.
    3. **Experiments with *smaller* step sizes.** The chosen step sizes (e.g. 5% of MNIST = 3000 images) are very large. Most active learning works present results for step sizes of 1, 10, 50, etc. (see the experiments in the references below). With such a large step size a random (or near random) acquisition can easily perform well because there are already enough examples to train a decent model. With smaller step sizes the choice of acquired examples becomes much more important for good model performance.
    4. The adaptive noise magnitude \eta. An ablation, showing the performance when using a constant value for the noise magnitude (e.g. a value that does not depend on \Delta\theta^{(i)}, should be added to verify that this is an important part of the proposed method.
    5. No runtime comparisons. In the text there are brief discussions of the runtime for the proposed method and some of the baselines, however there are no empirical results to go with the discussion. Runtime comparisons should ideally be provided for all baselines and the proposed method.

* **Connections to other work.** The proposed method seems very closely related to doing active learning with a linearised Laplace approximation of the BNN posterior (see Immer et al. and Daxberger et al. for good descriptions of the linearised Laplace approximation). However this is not discussed at all. I suspect that this link could provided a more concrete link to the predictive variance (the provided connection being somewhat weak in my opinion.)


## References

### Active Learning

Andreas Kirsch, Joost van Amersfoort, Yarin Gal:
BatchBALD: Efficient and Diverse Batch Acquisition for Deep Bayesian Active Learning. NeurIPS 2019: 7024-7035

Robert Pinsler, Jonathan Gordon, Eric T. Nalisnick, José Miguel Hernández-Lobato:
Bayesian Batch Active Learning as Sparse Subset Approximation. NeurIPS 2019: 6356-6367

Neil Houlsby, Ferenc Huszár, Zoubin Ghahramani, and Máté Lengyel. Bayesian active learning for classification and preference learning. arXiv preprint arXiv:1112.5745, 2011

### Linearised Laplace

Alexander Immer, Matthias Bauer, Vincent Fortuin, Gunnar Rätsch, Mohammad Emtiyaz Khan:
Scalable Marginal Likelihood Estimation for Model Selection in Deep Learning. ICML 2021: 4563-4573

Erik A. Daxberger, Eric T. Nalisnick, James Urquhart Allingham, Javier Antorán, José Miguel Hernández-Lobato:
Bayesian Deep Learning via Subnetwork Inference. ICML 2021: 2510-2521


**Summary Of The Paper:**

This paper provides a new method for solving the active learning problem, i.e. choosing from a set of unlabeled examples that should be labelled and added to the training set. The proposed method makes use of 'noise stability' of inputs–specifically if the model output corresponding to an input changes to a large extent when small amounts of noise are added to the model parameters, then this input should be labelled. The authors compare their method to a range of existing AL methods and draw theoretical connections to predictive variance reduction.

**Summary Of The Review:**

While the proposed method is simple to understand and solves an important problem, I do not think the paper is of the standards required for ICLR primarily due to the weak experimental evaluation and lack of clarity in the writing.

++++ Updated review ++++

The authors have provided many clarifications and additional experimental results which do strengthen the paper. As a result, I have increased my score from 3 to 5. Unfortunately, the experimental evaluation is still a little weak which prevents me from increasing my score further. Similarly, I am not sure that the updated presentation is quite clear enough.

---

> ### Author Response · Authors · 2021-11-23
> **Thank you for your valuable comments**
>
> We thank you for your constructive comments. To address your concerns, we provide some clarifications below. We may have not explained several points clearly enough in the manuscript. So, please feel free to raise any questions during the discussion phase and we will be happy to answer that.
>
>
>
> **Clarity**
>
> 1. Semi-supervised learning requires a certain amount of labeled data, which can be obtained from AL.
>
> 2. We made a mistake in writing. "representative" should be replaced by "informative".
>
> 3. Yes, "interlayer cushion" is a specific term used in Arora et al., 2018. We replace the term with a more general description for easy understanding.
>
> 4. The sentence is to explain that, models which are robust to input perturbations tend to be easy to recognize future (has not been observed in the existing training set) examples. We replaced “future” with “unseen” to avoid potential misleading.
>
> 5. We changed “by computing the distance between outputs of the clear and perturbed input” to “measuring how far is the output deviated from the original value”
>
> 6. We provided some examples at the second paragraph of **Introduction**
>
> 7. We follow the terminologies used in Yoo and Kweon, 2019 (LL4AL, see 5th paragraph in their introduction part) and Caramalau, 2021 (GCN, see abstract and section 3.2).
>
> 8. The same terminology is also used in the two papers mentioned above.
>
> 9. We agree that the set update equations can be removed for simplicity when describing the algorithm. We keep the formulation of the set update to help readers understand the experimental settings (iteratively adding newly annotated examples)
>
> 10. We revised Section 3.2 to keep the notation of Jacobian consistent.
>
> 11. Yes, we fixed it.
>
> 12. Eq 7 comes from Thisted, 1988 and is used by multiple followed studies. We will add a brief introduction in the appendix in the next version.
>
> 13. No, we only replace $J_\hat{y}(\tilde{x})^T$ by $u(\tilde{x})$.
>
>
> **Experimental evaluation**
>
> **1. The narrow range of underlying tasks.**
>
> We have evaluated our method on the 3D cryo-ET dataset (Now moved to Appendix A4.1), which is not a standard computer vision task. The input of cryo-ET is not an RGB image. In the revised manuscript, we provided experiments on a regression task in Appendix A4.2. Our method outperforms the SOTA baselines.
>
> **2. Missing baselines**
>
> As there are several mainstream ideas to perform uncertainty estimation, we only select 1-2 typical methods for each type of ideas from recent highly cited publications (mainly in 2016-2021). These baselines cover different aspects including deep Bayesian AL (MC-Dropout), batch-aware acquisition (CoreSet), version space (Query-by-committee), adversarial learning (VAAL, SRAAL) and special uncertainty estimation modules (LL4AL, GCN). However, we agree that comparing more deep Bayesian AL methods is helpful to verify the effectiveness of our method, as our method is mostly relevant to this category of methods. We provided additional experimental results on MNIST and CIFAR-10 in Appendix A.3, comparing our method with BALD, BatchBALD and Ensemble. Due to the resource limitation, we will add the experiment for Sparse Subset Approximation in the next version.
>
> **3. Experiments with *smaller* step sizes.**
>
> We added experiments with smaller step sizes 1/10/50 on MNIST. Results in Appendix A.4.3 show that our method performs well with smaller step sizes.
>
> **4. The effect of adaptive noise magnitude $\eta$.**
>
> We added an ablation study in Appendix A.5.1. Results show that the adaptive noise magnitude is reasonable and necessary. Using a constant magnitude of noise usually performs worse than the adaptive strategy. Moreover, the proper absolute magnitude relies on the parameter magnitude of a specific model. In contrast, the adaptive strategy is model-agnostic.
>
> **5. No runtime comparisons.**
>
> We provided the runtime comparison in Appendix A.6. Our method is relatively efficient among various AL baselines.
>
>
> **Connections to other work.**
>
> Thanks for recommending the references. Yes, they are very relevant and we add the discussions of the connections and differences in the revision (Appendix A.8).

---

> > ### Comment · Reviewer_owNu · 2021-11-25
> > **Score increase**
> >
> > Thank you for the detailed response, including many clarifications and additional experimental results.
> >
> > Regarding the narrow range of underlying tasks, I still feel that this is a weakness of the paper. While the new regression experiment certainly helps, the fact that cryo-ET is not RGB images is not a big enough differentiator from the other computer vision tasks in the experiments, in my opinion. I am also still skeptical that the majority of the experiments in the paper are done with very large batch acquisition sizes. Smaller sizes are where active learning really matters, so this is where the paper should be focused.
> >
> > Nevertheless, the additional experimental work does make this submission stronger. As a result, I have increased my score.

---

> > > ### Author Response · Authors · 2021-11-29
> > > **Thank you for increasing the score and for your valuable comments**
> > >
> > > Thank you for increasing the score and for your valuable comments.  Here we provide additional explanations about our experimental designs.
> > >
> > > 1. Regarding the range of underlying tasks, one fact is that most state-of-the-art AL methods (e.g. QBC, LL4AL, VAAL, SRAAL, GCN, CoreSet[1], MC-Dropout[2], BatchBALD[3]) are only evaluated on computer vision challenges. Some of them [1, 2, 3] are only validated on image classification tasks, while we validate our proposed method on both image classification and semantic segmentation. We agree that it would be more convincing to validate the method on different domains, thus we provide an experimental analysis on NLP data below. We will add this analysis in a future revised version.
> > > In order to verify the domain-agnostic nature of our method, we conduct an experiment on MRPC, which is a classical NLP task aiming to identify if two sentences are paraphrases of each other. At each AL cycle, we fine-tune the standard BERT (bert-base-uncased) model for 10 epochs to ensure convergence. Other hyper-parameters are determined following the official guide of Pytorch-Transformers-HuggingFace. We report the results in Table 3. As can be seen, our method delivers very promising results, demonstrating its effectiveness on various domains.  BALD is also competitive on this task, but it is not applicable to regression problems.
> > >
> > > Table 3. Accuracy (%) on the Microsoft Research Paraphrase Corpus (MRPC) dataset with 100 initial labeled samples for training. At each AL cycle, 50 additional samples are selected for annotation. For MC-Dropout/BALD/BatchBALD, we perform 50 feed-forward passes for Monte-Carlo sampling. For our NoiseStability, we still set the number of noise sampling to 10. All the reported results are averaged over 5 runs.
> > >
> > > |Method/Cycle| C0  | C1  | C2  | C3  | C4  | C5  | C6  |C7|C8|C9|
> > > |---|---|---|---|---|---|---|---|---|---|---|
> > > | Random  |       62.70|68.03|68.66|69.36|70.24|70.55|70.94|71.46|71.08|71.64 |
> > > | MC-Dropout | 62.70|69.12|69.56|70.15|70.78|71.32|71.47|72.21|72.94|74.17|
> > > | BALD  |           62.70|**69.71**|69.46|**70.78**|**71.96**|**72.75**|**73.04**|73.28|73.14|74.51|
> > > | BatchBALD  | 62.70|68.38|69.51|69.66|70.59|71.03|71.86|71.62|71.67|72.75|
> > > | CoreSet |        62.70|68.53|68.53|70.39|71.57|71.62|72.89|72.50|72.79|72.79|
> > > | GCN |             62.70|68.87|**69.71**|69.75|71.18|71.32|71.62|72.65|72.70|73.28|
> > > | NoiseStability |62.70|69.31|**69.75**|**70.74**|71.18|72.30|72.45|**73.48**|**73.58**|**74.61**|
> > >
> > >
> > > 2. As for the step size, the baseline AL methods have different attempts. MC-Dropout and BatchBALD mainly focus on small step sizes (e.g. less than 100), whereas CoreSet, LL4AL, VAAL and GCN focus on large step sizes (e.g. more than 1000). Sparse Subset Approximation [4] considers both. We argue that both small and large step sizes are important for evaluating AL methods. We agree that using small step sizes tends to yield a remarkable superiority of an AL method (e.g. AL with a step size of 1 will obtain the best performance). Nevertheless, it would be more practical to exploit a large step size in real-world scenarios since a small one will result in inefficiency. In the revised version, we used small step sizes for the MNIST and House Price Prediction tasks. It is worth noting that the existing works only reported the results of using small step sizes on simple datasets, such as MNIST or Housing Price Prediction, but they did not attempt to validate the small step sizes on complex datasets, such as CIFAR-10. This is mainly because small step sizes are inappropriate for such complex datasets, and the reasons are as follows. When the total number of labeled samples is small (e.g. in earlier AL cycles), with very few newly labeled samples added (due to small step sizes), the results obtained using different seeds (for multiple runs) show a high variance. Then it will be hard to tell which leads to the performance gain, newly added labeled data or randomness? When the total number of labeled samples is large (e.g. in later AL cycles), with very few newly labeled samples added (due to small step sizes), the performance gain is subject to be trivial for comparing the algorithms. Therefore, in this work we follow the same practice to only validate the small step sizes on simple datasets, such as MNIST and House Price Prediction. Nevertheless, we do agree that paying attention to small step sizes will make the work more solid. We will follow your valuable suggestions to further improve our work.
> > >
> > > [1] Active learning for convolutional neural networks: A core-set approach. (ICLR’18)
> > >
> > > [2] Deep Bayesian Active Learning with Image Data. (ICML’17)
> > >
> > > [3] BatchBALD- Efficient and Diverse Batch Acquisition for Deep Bayesian Active Learning (NeurIPS’19)
> > >
> > > [4] Bayesian Batch Active Learning as Sparse Subset Approximation. (NeurIPS’19)

---

### Author Response · Authors · 2021-11-23
**General response to all the reviewers**



We appreciate all the reviewers' recognition of the novelty, simplicity, as well as practicality of the proposed method.


Since most of the reviewers' questions lie in concerning the insufficient experiments, we add many new experimental results and the analysis in the revised version. Below we briefly introduce the newly added experiments and we refer the reviewers to our revised manuscript for details.

1. We add a comparison with the typical deep Bayesian active learning methods (Appendix A.3).

2. We correct the implementation of the MC-Dropout method, and observe that our method can outperform MC-Dropout even though our method needs much fewer feed-forward passes (see all the figures which involve MC-Dropout).

3. We add a regression experiment on a benchmark dataset (Appendix A.4.2).

4. We demonstrate the superiority of our method when small selection sizes (e.g. 1, 10, and 50) are adopted (Appendix A.4.3).

5. We validate our method on ImageNet, demonstrating its effectiveness on very large scale data (Appendix A.4.4).

6. We demonstrate the architecture-agnostic nature of our method by validating it with MobileNet, which is a quite different architecture from ResNet-18 that we used in the original manuscript (Appendix A.4.5).

7. We add a new ablation study to investigate the impact of *adaptive noise magnitude* and *position of noise injection*, respectively (Appendix A.5).

8. We add the running time comparison for different acquisition functions (Appendix A.6).

9. We add a detailed discussion about the relevant work about linearised Laplace approximation of the BNN posterior (Appendix A.8).


In addition, we also fix several presentation issues, including typos. We will continue improving the manuscript to make the writing more friendly to readers.

---

### Decision · Program_Chairs · 2022-01-20

**Decision:**

Reject

**Comment:**

The submission considers a new acquisition function for active learning. The method considers the sensitivity of the prediction for a given datapoint with respect to parameter perturbations. Points with the largest variance under these perturbations are selected for labelling.  The method is simple and the empirical results are reasonable. Some weaknesses are the clarity of writing, and somewhat limited experimental comparisons.

The discussion was useful and helped improve the clarity. Additional experiments also helped improve the paper, although some reviewers still felt the experimental comparisons were lacking, including using entropy as a baseline acquisition function. Despite these improved scores, the overall average score remains below threshold I'm afraid.

I feel this is a useful paper, but perhaps needs a little more polishing in the writing and some additional experiments. As such, it just falls short of the acceptance threshold.